# Comparative Metabolite Profiling Between *Cordyceps sinensis* and Other *Cordyceps* by Untargeted UHPLC-MS/MS

**DOI:** 10.3390/biology14020118

**Published:** 2025-01-23

**Authors:** Jing Ma, Zhenjiang Chen, Kamran Malik, Chunjie Li

**Affiliations:** 1Grassland Research Center of National Forestry and Grassland Administration, Institute of Ecological Protection and Restoration, Chinese Academy of Forestry, Beijing 100091, China; mjtwoseven@163.com; 2State Key Laboratory of Herbage Improvement and Grassland Agro-Ecosystems, Key Laboratory of Grassland Livestock Industry Innovation (Ministry of Agriculture and Rural Affairs), Engineering Research Center of Grassland Industry (Ministry of Education), Gansu Tech Innovation Centre of Western China Grassland Industry, Center for Grassland Microbiome, College of Pastoral Agriculture Science and Technology, Lanzhou University, Lanzhou 730020, China; chenzhenjiang@lzu.edu.cn (Z.C.); malik@lzu.edu.cn (K.M.)

**Keywords:** *Cordyceps sinensis*, LC-MS, metabolomics, counterfeit, identify

## Abstract

*Cordyceps sinensis* plays an important role in treating diseases and providing physiological healthcare, but its growth and quantity are scarce. Our study compared the differential metabolites between *C. sinensis* and other *Cordyceps* using UHPLC-MS/MS. Our research indicates that there are significant differences between *C. sinensis* and *C. liangshanensis*, *C. gracilis*, *C. hawkesii*, and *C. gunnii*. Nucleotides and nucleosides, amino acids and derivatives, lipids and lipid molecules, and organic acids and their derivatives were the key metabolites that distinguished *C. sinensis* from other groups. The types and abundance of nucleoside differential metabolites in *C. sinensis* are superior to other Cordyceps. Amino acid metabolism and lipid metabolism were the metabolic pathways with the greatest differences between *C. sinensis* and the other four Cordyceps. These research results provide a theoretical basis for distinguishing between *C. sinensis* and mixed products, and also provide a reference for the development and utilization of *Cordyceps* for healthcare in the future.

## 1. Introduction

*Cordyceps* is a complex of larval corpses and fungal spores formed by the infection of the genus *Cordyceps* of the family Ophiocordycipitaceae in the larvae of insects and other arthropods [1]. More than 600 species of wild *Cordyceps* have been discovered worldwide. Among them, there are about 350 species of *Cordyceps* in China, including *Cordyceps sinensis, Cordyceps liangshanensis*, *Cordyceps gunnii*, *Cordyceps gracilis,* and *Cordyceps hawkesii* [2]. *C. sinensis*, called Chinese *Cordyceps* or DongChongXiaCao, is the most widely sold and recognized. *C. sinensis* mainly grows at altitudes between 3000 and 5000 m in shrubs and meadows in Qinghai, Gansu, Tibet, Yunnan, and other provinces and regions in China [3]. The significant differences in geographical location result in significant variations in *Cordyceps* in different growth environments, ultimately manifested as differences in metabolites [4]. The quality of C. sinensis produced in Naqu (Tibet) and Yushu (Qinghai) is better than other places, with Yushu being particularly renowned as the “hometown of *C. sinensis*”. The production of *C. sinensis* in Yushu reaches 45 tons, accounting for 20% of the total national production, with an annual output value of 7 billion yuan, accounting for 54.6% of the net income of the Yushu people in 2023. The research hotspots of *C. sinensis* focus on chemical composition, medicinal value, and mycelial fermentation, and little research has been devoted to identifying different types of *Cordyceps* through different metabolic components. Due to geographical differences, there are differences in metabolites among *Cordyceps* in different growth areas [5].

*C. sinensis* has been used as a precious traditional Chinese medicine for thousands of years. Initial records of *Cordyceps* as medicine appeared in Wu Yiluo’s work *Ben-Cao-Cong-Xin* in the Qing Dynasty around 1757 [6]. Therefore, *C. sinensis* has been used as a health supplement and tonic for subhealth patients. There are over 350 species of *Cordyceps* that have been reported globally, but only a few of them can be used for health products and therapeutic drugs [7]. *C. sinensis* contains a variety of metabolites such as amino acids, pantothenic acid, mannitol, sterols, and polysaccharides [8,9,10,11]. Some studies have demonstrated that nucleosides, polysaccharides, and some amino acids have important pharmacological activities in *C. sinensis* [12,13]. *C. sinensis* has strong reducing power and can eliminate free radicals both inside and outside the body. Studies have shown that this is related to the rich variety of amino acids in *C. sinensis* [14]. *C. sinensis* has several properties, such as anti-oxidation, immunoregulatory, cholesterol-lowering, antithrombotic, hypolipidemic, hypoglycemic, and vasodilation properties. Additionally, it has depressive effects and can enhance physical fitness and longevity [15]. Therefore, *C. sinensis* has been used as a health supplement and tonic for subhealth patients. Adenosine is commonly considered a marker bioactive component in *Cordyceps*-related species [16]. Research has shown that adenosine in *Cordyceps* has significant anti-inflammatory, anti-fatigue, anti-tumor, anti-aging, and other special pharmacological effects [17,18]. The combination of these effects has led to the widespread research and application of *Cordyceps* in the fields of healthcare and treatment [19,20]. *Cordyceps* is used as an energy supplement for athletes. The research team found through studying mice that *Cordyceps* helps to clear lactate and enhance anaerobic respiration in mouse cells [21]. On the other hand, energy and improving internal mechanisms can increase the concentration of cellular bioenergy ATP, thereby improving the efficiency of oxygen utilization and enabling athletes to compete in oxygen-scarce conditions [22]. Therefore, comparing the differential metabolites and screening potential biomarker compounds is one of the urgent problems to be solved.

Metabolomics is an emerging discipline that has developed after genomics and proteomics. It is a high-throughput, ultra-sensitive, and wide-ranging technique that can be used for qualitative and quantitative analysis [23]. One of the mass spectrometry (MS) techniques, called LC-MS, can detect substances which are difficult to volatilize and unstable metabolites, with a wider range of applications [24]. However, few studies have employed untargeted LC-MS metabolomics for *Cordyceps* identification [25]. Currently, a variety of other *Cordyceps* have increasingly appeared on the market, including mixed products like *C. hawkesii*, *C. militaris*, *C. liangshanensis*, and *C. gracilis,* which are challenging to differentiate solely by appearance [2]. Research has shown that these species differ significantly in both appearance and efficacy compared to *C. sinensis* [26]. In response to the issue of counterfeit and substandard products, several methods were proposed to authenticate *C. sinensis* [27]. As research deepens, we need other compounds to distinguish *C. sinensis* from other *Cordyceps*, and further study the differences in medicinal functions of different *Cordyceps*. In this study, we used untargeted LC-MS metabolomics to compare key differential metabolites between *C. sinensis* and *C. gracilis*, *C. hawkesii*, *C. liangshanensis,* and *C. gunnii*, offering a novel approach for distinguishing and identifying *C. sinensis* and its substitutes. Overall, our study aims to explain the differences between *C. sinensis* and other *Cordyceps* in a complete set of metabolites, providing a scientific basis for the identification and screening of biomarkers of *Cordyceps*.

## 2. Materials and Methods

### 2.1. Materials and Reagents

*C. sinensis*, *C. liangshanensis*, *C. gracilis*, *C. hawkesii*, and *C. gunnii* were collected and identified authentic strains by Professor Li Yuling’s *Cordyceps* research team, which are stored in the Microbial Center strain storage library [5,28]. Different specimens of mature *Cordyceps* are shown in Figure 1. Reagents such as methanol, acetonitrile, formic acid, pure water, and propanol were all purchased from Thermo Fisher Scientific (Waltham, MA, USA).

### 2.2. Samples Collection

We selected 5 *Cordyceps* at the same maturity stage from various production areas and set up 6 biological repeats. The samples were cleaned with distilled water and air-dried. Then, a 50 mg solid sample from each *Cordyceps* was added in a 2 mL centrifuge tube sample, as well as a 6 mm steel ball. We added 400 μL of extraction solution (methanol–water = 4:1, *v*/*v*), which contained 0.02 mg/mL internal standard (L-2-chlorophenylalanine) [28]. The sample solution was ground by the Wonbio-96c (Shanghai Wanbo Biotechnology Co., Ltd., Shanghai, China) frozen tissue grinder for 6 min (−10 °C, 50 Hz), followed by low-temperature ultrasound extraction for 30 min (5 °C, 40 kHz). The samples were placed at −20 °C for 30 min, then centrifuged at 12,000 rpm for 10 min (4 °C). The supernatant was collected, filtered, and loaded into an automatic sampler vial for LC-MS analysis. The samples were tested by Shanghai Meiji Biopharmaceutical Technology Co (Shanghai, China).

### 2.3. Quality Control Samples

Quality control samples (QC), as part of the system tuning and quality control process, were processed by mixing all test samples of the same volume and then using the same pre-treatment method as the test samples. It helped to represent the entire sample set, which was regularly injected to monitor the stability and reliability of the analysis.

### 2.4. UHPLC-MS/MS Analysis

The LC-MS/MS analysis of the samples was performed on an ultra-high-performance liquid chromatography system, Thermo UHPLC-Q Exactive HF-X (Thermo Scientific, Waltham, MA, USA), which was equipped with ACQUITY HSS T3 columns (100 mm × 2.1 mm i.d., 1.8 μm, Waters, Milford, MA, USA) at Majorbio Bio Pharm Technology Co., Ltd. (Shanghai, China). The mobile phase consists of a solution of 0.1% formic acid in water (solvent A) and a mixture of 0.1% formic acid in acetonitrile (solvent B). The positive-ion-mode separation gradient was set as follows: 0–3 min, 20% B; 3–4.5 min, 20–35% B; 4.5–5 min, 35–100% B; 5–6 min, 100% B; 6–6.5 min, 100–0% B; 6.5–8 min, 0% B. The negative-ion-mode separation gradient was set as follows: 0–2 min, 0–10% B; 2–4.5 min, 10–30% B; 4.5–5 min, 30–100% B; 5–6 min, 100% B; 6–6.5 min, 100–0% B; 6.5–8 min, 0% B. The column temperature was 40 °C with a flow rate of 0.2 mL/min and an injection volume of 3 µL. The column temperature was 40 °C and the flow rate was 0.40 mL/min.

Sample mass spectrometry signals were collected using a Thermo UHPLC-Q Exactive HF-X Mass Spectrometer equipped with an electrospray ionization (ESI) source operating in positive mode and negative mode, with a mass scanning range of 70–1050 m/z. The optimal process conditions were: source temperature at 425 °C; Aux gas flow rate at 13 arb; sheath gas flow rate at 50 arb; ion-spray voltage floating (ISVF) at −4500 V in negative mode and 4500 V in positive mode, respectively; normalized collision energy, 20–40–60 V rolling for MS/MS. The full MS resolution was 60,000, and the MS/MS resolution was 7500. Data acquisition was performed with the Data-Dependent Acquisition (DDA) mode. The detection was carried out over a mass range of 70–1050 m/z.

### 2.5. Data Anlaysis

The pre-processing of LC/MS raw data was carried out using Progenesis QI (version 3.0) (Waters Corporation, Milford, MA, USA) software for baseline filtering, peak identification, integration, retention time correction, and peak alignment, which were carried out in the data matrix. Internal standard peaks and any known false positive peaks were removed from the data matrix along with redundancy and merging peaks. At the same time, metabolic products were identified by searching databases, with the main database being the Human Metabolome Database (HMDB, http://www.hmdb.ca/ (accessed on 23 April 2024)), Metlin (https://metlin.scripps.edu/ (accessed on 23 April 2024)), and Majorbio databases. The data matrix obtained by searching the databases was uploaded to the Majorbio cloud platform (https://cloud.majorbio.com (accessed on 15 May 2024)) for data analysis. The data matrix was pre-processed, and at least 80% of the metabolic features detected in any set of samples were retained. After filtering, the minimum metabolite value was estimated and each metabolic signature was normalized to the sum. To mitigate errors arising from sample preparation and instrument instability, the response intensities of the sample mass spectrometry peaks underwent normalization using the sum normalization method, resulting in the creation of a normalized data matrix. Additionally, variables from QC samples exhibiting a relative standard deviation (RSD) > 30% were excluded and logarithmically transformed using log10, yielding the final data matrix utilized for subsequent analysis.

Principal Component Analysis (PCA) was performed by the ropls package in R language (Version 1.6.2). PCA transformed multiple indicators into a few comprehensive indicators through dimensionality reduction, determined the correlation between variables, and increased the complexity of analysis. Differential metabolites were annotated through metabolic pathways in the KEGG database (https://www.kegg.jp/kegg/ (accessed on 18 May 2024)). We performed a pathway enrichment analysis using the Python package (version 3.8) “scipy.stats” (https://docs.scipy.org/doc/scipy/ (accessed on 20 May 2024)). Enrichment analysis categorized differential metabolites based on their involvement in pathways and functions, thereby analyzing sample diversity. An analysis of metabolic component data of *Cordyceps* was performed using Majorbio cloud platform (cloud.majorbio.com (accessed on 24 May 2024)).

## 3. Results

### 3.1. Principal Component Analysis of C. sinensis and Other Cordyceps

We use UHPLC-Q Exactive HF-X untargeted metabolomics to obtain characteristic differential metabolites of *C. gracilis* (XJ), *C. sinensis* (YS), *C. hawkesii* (YX), *C. liangshanensis* (LS), and *C. gunnii* (GN). PCA determines the stability of the instrument and the validity of the data by analyzing the repeatability and dispersion of differences in the sample. A smaller dispersion indicates greater instrument stability and more reliable test results. All samples were within the 95% confidence interval, and the biological replicates of YS were very concentrated (Figure 2), proving that the data were reproducible and reliable. The results of the PCA (Figure 2a) revealed that PC1 had a contribution rate of 48.20%, while PC2 had a contribution rate of 12.10%, with the cumulative contribution rate of PC1 and PC2 reaching 60.30%, which indicated a clean separation of the samples by the components. The overlapping components of YS and YX indicated the presence of identical metabolic components in the sample. Meanwhile, YX duplicate samples exhibited high dispersion and significant differences in metabolic components. For the metabolomics profile, good separation was achieved between YS and the GN, LS, and XJ samples based on the overlapping components in the Principal Component Analysis (Figure 2b–d).

*C. gracilis*, *C. sinensis*, *C. hawkesii*, *C. liangshanensis*, and *C. gunnii* are represented by XJ, YS, YX, LS, and GN, respectively, and are consistent throughout the text.

### 3.2. Comparative Analysis of Metabolites

We detected and identified 4898 metabolites from 18 superclass in the sample, and found that the chemical constituents of THE samples included lipids, amino acids, nucleotides, carbohydrates, organic acids, vitamins, and their derivatives or other small molecules (Appendix A). Appendix A lists 1257 common metabolic components among the five *Cordyceps*. Importantly, some widely used bioactive components or biomarkers were identified in all samples, such as nucleosides (adenosine), alcohols (ergothioneine and panthenol), and amino acids (histidine, glutamic acid, and proline). Appendix A showed 31 specific metabolic components and their HMDB IDs of YS, which can be used as a special metabolite to distinguish *C. sinensis* from the other four types of *Cordyceps*. Metabolites such as glutamic acid [19], 3′- thiocytidine, and stearic acid were not detected in GN, LS, XJ, and YX samples. We detected the differential metabolites of anions and cations between YS and XJ and other *Cordyceps* using ion-source mode, and the results showed that there was not much difference in the number of differential metabolites of anions and cations between the groups (Table 1). However, only a limited number of metabolites could be identified by name among the detected cations and anions, while the rest remained unknown, highlighting the need for further investigation.

Figure 3a showed that there were 789 overlapped differential metabolites (including lipids, amino acids, and nucleotides) between YS and GN, and 1488 overlapped differential metabolites between YS and LS in Figure 3b. Compared with YS, there were 1579 and 1712 overlapping differential metabolites in XJ and YX samples, respectively (Figure 3c,d). There were more differential metabolites (including nucleotides, amino acids, and lipids) between the YS and XJ *Cordyceps* species than between YS and other *Cordyceps* species. In summary, we detected a large number of various metabolites in the *Cordyceps* samples, which could help to comprehensively evaluate their differences. Appendix A, respectively, showed the unique chemical components of YS samples compared to the GN, LS, XJ, and YX samples, including lipids, nucleotides, carbohydrates, organic acids, and other small molecules. It is worth noting that more lipids, organic acids, and their derivatives were detected in the differential metabolites. These metabolites are also potential biomarkers for distinguishing *C. sinensis* from the other four *Cordyceps* species.

The red color represented the upregulated differential metabolites, while blue represented the downregulated differential metabolites. Statistical analysis and a comparison were conducted pairwise on the differential metabolites of *C. sinensis* (VIP ≥ 1, FC ≥ 2 or ≤ 0.5); the results showed that 1769 significantly altered metabolites (618 upregulated, 1151 downregulated) were found between YS and GN (Figure 3e), and 1596 significantly altered metabolites (956 upregulated, 640 downregulated) were found between YS and LS (Figure 3f). As shown in Figure 3g, a total of 1878 differential metabolic components were identified in XJ compared with YS, of which 797 and 1081 differential metabolic components were upregulated and downregulated, respectively. Compared with YS, there were a total of 1597 differential metabolic components in the YX sample, of which 397 were significantly upregulated while 1199 were significantly downregulated (Figure 3h).

Figure 3 showed that the metabolites of YS significantly differed from other *Cordyceps*, mainly concentrated in lipids, organic acids, organic oxides, and nucleosides. The results revealed that there were certain metabolic differences in wild *Cordyceps* from different regions, and unique ecological factors in the local area might affect the chemical composition of wild *Cordyceps*.

It is worth noting that some compounds in the “other” category, as well as some compounds such as proscillidin and glutethimide, may originate from consumable materials and experimental operations that are considered pollutants, rather than the natural volatile compounds of *Cordyceps* [29]. Therefore, most of these compounds are excluded from further analysis and discussion.

### 3.3. Screening for Significantly Different Metabolites

A comparative analysis was conducted on the VIP values of differential metabolites between YS and other *Cordyceps* samples, and nine metabolites with significant differences were selected. These differential metabolites were mainly concentrated in lipids and lipid-like molecules, organic acids and derivatives, organic heterocyclic compounds, and phenolpropanoids and polyketides. Zedoarol, hyloglycin B, P-Cttlvc-T, 6a-hydrox-ypaclitaxel, and PA (20:3 (8Z, 11Z, 14Z) -2OH (5,6)/i-12:0) were found to have a significantly higher abundance in YS than in the other four samples, while less (+)—Isobridgene, sarporgrelate, and proscillaridin were found in YS (Figure 4). It could be speculated that these metabolites might be influenced by environmental factors. In summary, the differential metabolites of *Cordyceps* are mainly concentrated in lipids and organic acids, providing us with a new direction for screening suitable biomarkers between *C. sinensis* and four easily confused products.

### 3.4. Relative Quantitative Analysis of Nucleotide-Related Metabolites in C. sinensis and Its Adulterants

Many reported active ingredients of *Cordyceps* are related to the presence of nucleoside compounds. Targeted relative quantitative analyses showed that didanosine, 5-Methyldioxycytidine, and N6-Methyl-2′-deoxyadenosine had a high level in YS (Figure 5A). Meanwhile, the content of didanosine and cyclohexyladenosine was significantly higher in YS and YX than that in other three types of *Cordyceps*. A total of 14 differentially metabolized nucleoside compounds were identified as given in Appendix A, and the percentage of nucleosides in each type of *Cordyceps* varies (Figure 5B). The results showed that the types of nucleoside compounds in *C. sinensis* were significantly more abundant than those in other *Cordyceps*. The relative content of 5-Methyldioxycytidine was in the order of YS > YX > GN > XJ = LS = 0, the relative content of cyclohexy ladenosine was YS > YX > GN > LS > XJ, and the proportion of content in XJ was 0, which was used to determine the relationship between XJ and YS. Didanosine was significantly present in YS, 2′, and 3′—Deoxy-3′- fluorouridine, which had the highest content in GN. Cytidine and (S) -5′- Deoxy-5′—(methylsulfonyl) adenosine were only absent in LS and were used to distinguish it from YS.

### 3.5. Partial Differential Metabolic Pathway Analysis

As shown in Figure 4, KEGG enrichment analysis showed significant differences in metabolic pathways among the top 20. Different metabolic pathways are the main reason for the differences in *Cordyceps* production. Through KEGG metabolic pathway enrichment analysis, the differential metabolites between YS and other *Cordyceps* were further studied. The results showed that, compared with YS, “arginine and proline metabolism”, “sphingophospholipid metabolism”, and “glycerophospholipid metabolism” were the top three significantly enriched pathways with the lowest *p*-value in the fruit body in the metabolic category (Figure 6a). The differential metabolites in YS and LS were mainly enriched with “glycerophospholipid metabolism”, “nucleotide metabolism”, and “linoleic acid metabolism” (Figure 6b). The differential metabolism between YS and XJ samples was mainly concentrated in “arachidonic acid metabolism”, “tyrosine metabolism”, and “glycerophospholipid metabolism” (Figure 6c). The top three significantly enriched pathways with the lowest *p* values in YS and YX fruits were “arginine and proline metabolism”, “sphingophospholipid metabolism”, and “glycerophospholipid metabolism” (Figure 6d). It is worth noting that the first three significantly enriched metabolic pathways between the YS and GN samples were consistent with the metabolic pathways between the YS and YX samples.

## 4. Discussion

As both a medicinal and health product, *C. sinensis* has a long history in China, spanning thousands of years, and is widely valued by locals. Previous studies have shown that the metabolites of *C. sinensis* often determine its medicinal value [8]. *Cordyceps* growing in different environments may undergo significant metabolite changes [14,30,31,32]. Tan et al. found that *C. liangshanensis* has a few similarities with *C. sinensis*, but it differs in chemical composition [33]. Studies suggested that *C. sinensis* could be easily confused with the products of *C. gracilis*, *C. hawkesii*, *C. liangshanensis*, and *C. gunnii*. This experiment used untargeted metabolomics combined with multivariate data analysis methods to explore the differences in metabolites between *C. sinensis* and other *Cordyceps*. Untargeted metabolomics is a method that detects and analyzes all small-molecule metabolites in biological samples without bias. Therefore, the number of metabolites detected by untargeted metabolomics is higher than that detected by broadly targeted metabolomics [8,34]. Secondly, the extraction of metabolites is a crucial part of metabolomics research, as it directly affects the range of detectable metabolites and the quantity of metabolites extracted. The extraction solution used in this study is methanol–water = 4:1 (v: v), and the feasibility of this extraction method has been verified [28].

The current study determined a total of 4895 metabolites of 5 *Cordyceps* were detected in three databases, namely HMDB (http://www.hmdb.ca/ (accessed on 23 April 2024)), Metlin (https://metlin.scripps.edu/ (accessed on 23 April 2024)), and Majorbio. Among them, there were 3638 differential metabolites, mainly concentrated in amino acids and derivatives, carboxylic acids and derivatives, nucleotides and nucleosides, and glycerophospholipids. The metabolites of positive ions were more than negative ions. These metabolites were classified into 18 subclasses based on their functional similarity. Among them, the most abundant superclass included lipids and lipid-like molecules, organic acids and derivatives, organoheterocystic compounds, and benzenoids. These research findings were consistent with the reported studies [26,35,36]. The PCA and veen analysis showed a significant difference in the variation between *C. sinensis* and other *Cordyceps*, indicating that the metabolites of the same species were influenced by their growth environment. Most of the differential metabolites belonged to lipids and lipid-like molecules, as well as organic acids and derivatives. Compared with *C. hawkesii*, *C. gunnii, C. liangshanensis* and *C. gracilis*, eight significantly different metabolites were screened from the differential metabolites of *C. sinensis* (Figure 4), which served as potential biomarkers for quality evaluation and metabolite annotation in *Cordyceps* identification. The two types of metabolites, including organic acids and derivatives, and lipids and lipid molecules, showed the greatest relative abundance differences between *C. sinensis* and other *Cordyceps*, and were influenced by the growth environment of *Cordyceps* [37]. Among these compounds, 6a-hydrox-ypaclitaxel exhibits the ability to scavenge hydroxyl radicals and possesses strong reducing properties [38]. Its high concentration may be attributed to the abundance of amino acids, such as glutamic acid and arginine in *C. sinensis*, both of which are known for their strong reducing capabilities [38]. The content of two organic acids and their derivatives, namely hyogolycin B and P-Cttlvc-T, was significantly higher than that of the other four groups. According to previous studies, volatile compounds such as organic acids and their derivatives, organic heterocyclic compounds, and organic oxygen compounds can be easily affected by the environment [39]. Nitrogenous bases and nucleosides are important chemical markers for testing the quality of *C. sinensis,* such as nucleosides, adenosine, and guanosine [40,41]. Nucleosides have significant differences in type and content among different *Cordyceps* [42,43]. The differential analysis of metabolites showed that there were a total of 13 nucleosides, nucleotides, and analogs, with significant changes in the content of *C. sinensis* and other four types of *Cordyceps* samples. There were more types and contents of nucleoside metabolites in *C. sinensis*, while other *Cordyceps* were mostly concentrated on one or two nucleosides (Figure 5B). The current study also conducted KEGG metabolic pathway analysis to reveal the differential metabolic pathways between *C. sinensis* and other *Cordyceps*. The analysis revealed that these pathways were primarily concentrated in amino acid and lipid metabolism. Meanwhile, the pathways that showed significant differences in amino acid metabolism were the pathways of metabolism and synthesis of arginine and proline, as well as the metabolism of tyrosine, tryptophan, and lysine. Previous research has shown that regulating the metabolic pathways of arginine and proline is crucial in enhancing *Bactrocera dorsalis* resistance to low-temperature stress [44]. The content of arginine in *C. sinensis* was relatively high [8], which was consistent with our results. It indicated that temperature differences caused by regional differences could lead to changes in the resistance of *C. sinensis* to low temperatures, which was reflected in differences in the production of its metabolites. The differential metabolites were significantly enriched in the tyrosine metabolic pathway and tryptophan metabolism. We found 38 differential metabolites in the tryptophan metabolism pathway, as well as 27 differential metabolites in the lysine metabolism pathway. Studies have shown that amino acids are stress-responsive substances, such as tryptophan and lysine, which can significantly enhance the ability to resist stress. For instance, drought can promote the synthesis of tryptophan and lysine [36].

In addition, there were significant differences in the lipid metabolism pathways between *C. sinensis* and the other four groups of *Cordyceps*, including glycerophospholipid metabolism, linoleic acid metabolism, arachidonic acid metabolism, and sphingolipid metabolism. Lipid molecules are the fundamental components of living organisms and are influenced by the environment. Previous studies found that wild *C. sinensis* has a higher lipid content, which was consistent with the results of this study [35,45,46]. Many lipid metabolites also have immune activity. Savchenko et al. demonstrated that arachidonic acid can trigger plant defense and increase expression in general stress responses [47]. By analyzing the differential metabolic pathways of *C. sinensis* compared to other species, particularly in terms of amino acid metabolism, this study can provide valuable insights for the identification of genuine *C. sinensis* and its counterfeit products. Lipids and lipid molecules, due to their stable chemical structures, have different oxidation rates at different temperatures, forming short chains while promoting amino acid and carbohydrate metabolism [14,35,48]. The climate conditions and growth environment in the production area may be important factors affecting the lipid content of *Cordyceps* [24]. *C. sinensis* grown under high altitude, prolonged light exposure, and low-temperature conditions has higher lipids, which is consistent with our research results [5,28]. Zhang analyzed the changes in metabolites of *C. sinensis*, *C. cicadae*, and *C. gunnii*, and found that temperature was the largest influencing factor on nucleoside compounds [7]. As the altitude increases, there are plateau phenomena such as reduced precipitation, large temperature differences between day and night, and long sunshine hours [49]. Under these conditions, *C. sinensis* needs to produce more energy, accelerating the production of lipid metabolites [50].

Our study performed an in-depth analysis of the differential metabolites between *C. sinensis* and four other species of *Cordyceps*. Using untargeted LC-MS/MS, we found that the content and types of lipids, nucleosides, and other compounds in *C. sinensis* were significantly higher than other *Cordyceps*, establishing a solid scientific foundation for consumers to effectively distinguish *C. sinensis* from similar products. Given the absence of standardized criteria, non-targeted metabolomics analyses may yield a higher incidence of false positives. Furthermore, the broadly utilized non-targeted metabolomics techniques can inadvertently detect irrelevant substances [51]. Currently, these data have undergone pre-processing and normalization to enhance the accuracy of qualitative metabolomics results. However, relying solely on metabolomics to analyze the differential metabolites of *C. sinensis* and its substitutes has limitations. In future research, we will conduct a targeted metabolomic analysis of nucleoside compounds based on the results of this study, through accurate qualitative and quantitative analysis. We can analyze the five *Cordyceps* through a combination of targeted metabolomics and transcriptomics to understand their health benefits and potential value.

## 5. Conclusions

In summary, this study used untargeted LC-MS/MS metabolomics to analyze the metabolites of *C. sinensis* and its adulterants. We found there were significant differences in metabolites among *C. sinensis* and *C. hawkesii*, *C. gunnii, C. liangshanensis,* and *C. gracilis*. Nucleotides, lipids, organic acids, and their derivatives were the key metabolites that distinguish *C. sinensis* from other groups. However, the relative abundance of most bioactive nucleotides in *C. sinensis* was higher than that of the other four *Cordyceps*. There were significant differences in eight biomarkers, including Zedaorol, hylolycin B, P-Ctlvc-T, 6a-hydrox-ypaclitaxel, PA (20:3 (8Z, 11Z, 14Z) -2OH (5,6)/i-12:0), (+)—Isobridgene, sarporgrelate, and proscillaridin between *C. sinensis* and other alternatives. *C. sinensis* is rich in hydroxyinosine and is a potential HIV treatment drug. KEGG enrichment analysis showed that amino acid metabolism and lipid metabolism were the metabolic pathways with the greatest differences between *C. sinensis* and the other four *Cordyceps.* Overall, this study distinguishes differential metabolites between *C. sinensis* and its often-confused products, which not only provides chemical evidence for the identification of the five *Cordyceps*, but also facilitates better quality evaluations. Future efforts will focus on targeted metabolomics and transcriptomic analyses of these differential metabolites to further validate and quantify their production processes and pharmacological effects, providing a theoretical basis for the development and utilization of the pharmacological activities of *Cordyceps* and its substitutes.

## Figures and Tables

**Figure 1 biology-14-00118-f001:**
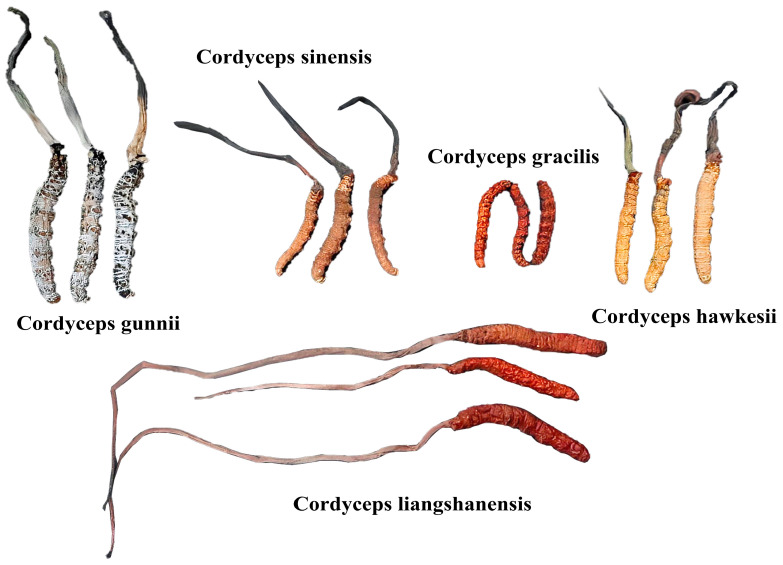
The specimens of mature *Cordyceps* of *C. gracilis*, *C. hawkesii*, *C. gunnii*, *C. sinensis* and *C. liangshanensis*.

**Figure 2 biology-14-00118-f002:**
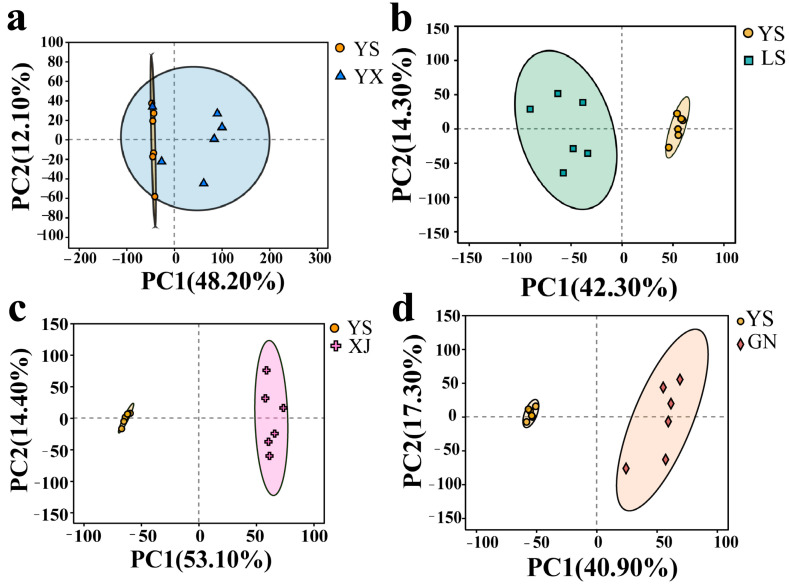
PCA results of total different metabolites between *C. sinensis* and *C. hawkesii* (**a**), *C. liangshanensis* (**b**), *C. gracilis* (**c**), *C.gunnii* (**d**).

**Figure 3 biology-14-00118-f003:**
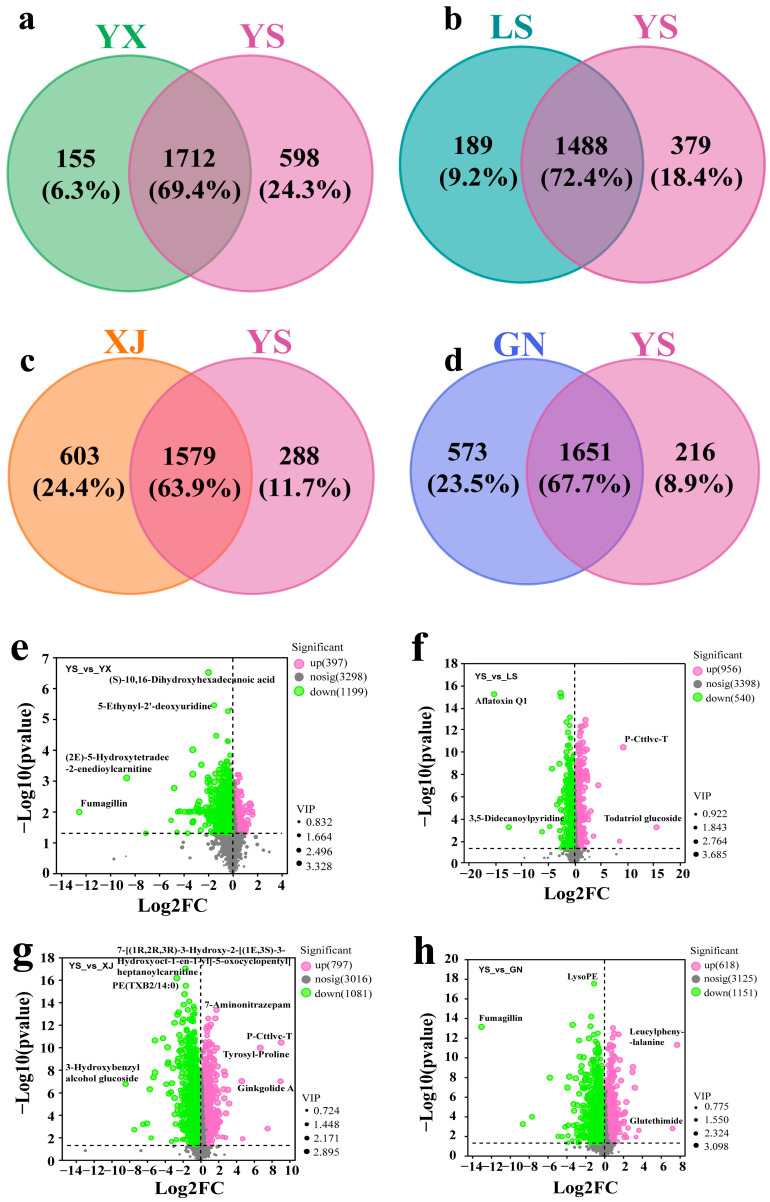
Analysis of differential metabolites in YS, YX, LS, XJ, and GN. The veen results of total different metabolites between YS and YS (**a**), LS (**b**), XJ (**c**), and GN (**d**). Volcanic map of total different metabolites between YS and YX (**e**), LS (**f**), XJ (**g**), and GN (**h**).

**Figure 4 biology-14-00118-f004:**
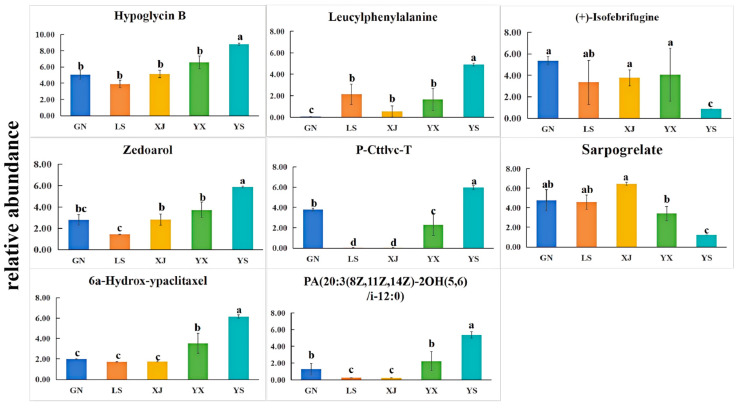
Significant differential metabolites among five types of *Cordyceps*. Relative abundance of nine metabolites with significant differential expression between YS and YX, LS, XJ, and GN. a–c Means without same supers within a column are significantly different (*p* < 0.01) Columns with the same letters are not significantly different. Different low case letters above columns indicate statistical differences at *p <* 0.05.

**Figure 5 biology-14-00118-f005:**
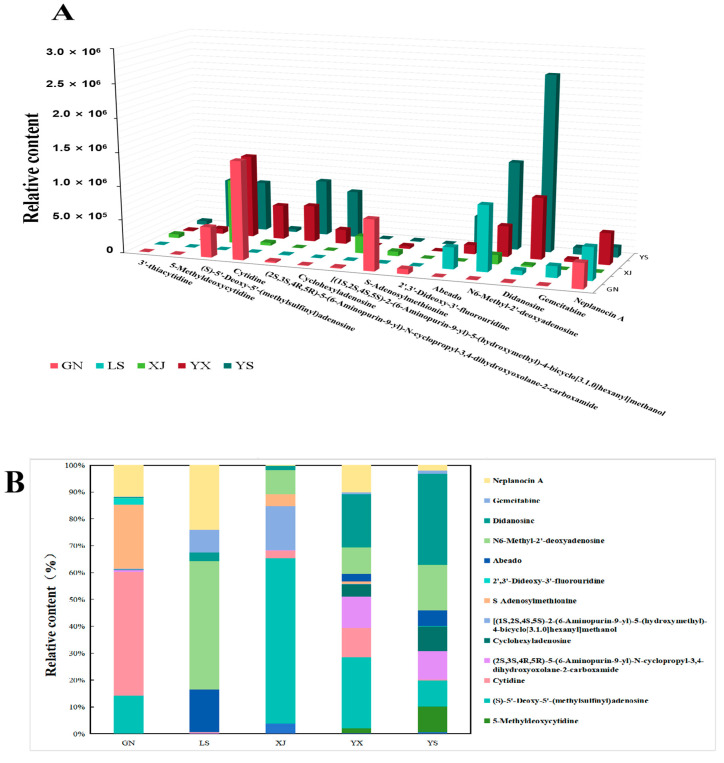
Differential nucleosides and their derivatives between YS and four other groups of *Cordyceps.* (**A**) Absolute quantitative results of some nucleosides and their derivatives; (**B**) a histogram of the relative content of major components in the nucleosides, nucleotides, and analog metabolites YS and YX, LS, XJ, and GN.

**Figure 6 biology-14-00118-f006:**
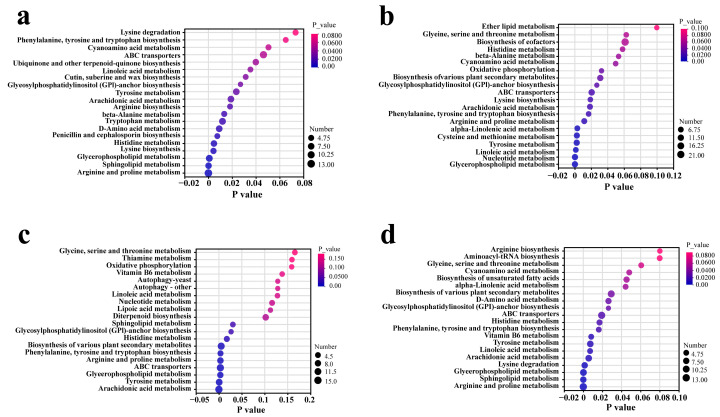
The top 20 KEGG enriched metabolic pathways (with *p*-values) are displayed. (**a**) Enrichment analysis of differential metabolites between YS and YX using the KEGG database. (**b**) Enrichment analysis of differential metabolites between YS and LS using the KEGG database. (**c**) Enrichment analysis of differential metabolites between YS and XJ using the KEGG database. (**d**) Enrichment analysis of differential metabolites between YS and GN using the KEGG database.

**Table 1 biology-14-00118-t001:** Differential metabolic statistics table.

Mode	Total Number	YS_vs_YX	YS_vs_XJ	YS_vs_GN	YS_vs_LS	XJ_vs_GN	XJ_vs_LS
pos	6122 (2107)	2259 (846)	2895 (998)	2607 (945)	2254 (768)	2845 (1063)	2820 (920)
neg	6181 (1905)	2341 (750)	2826 (880)	2538 (824)	2392 (728)	2624 (867)	2878 (878)

Note: The first column represents the ion source mode, the second column, Total Number, represents the union of the differential ion peaks of all differential groups, and the remaining columns represent the number of ion peaks included in each differential group. The number of ion peaks outside the parentheses that meet the differential screening criteria is all inclusive, while the number of named differential metabolites identified is indicated within the parentheses.

## Data Availability

The data that support the findings of this study are available from the corresponding author upon reasonable request.

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
