# Peer review of "Comparative Metabolite Profiling Between Cordyceps sinensis and Other Cordyceps by Untargeted UHPLC-MS/MS"

_biology, 2025, doi:10.3390/biology14020118_

Round 1

Reviewer 1 Report

Comments and Suggestions for Authors

Review of the research paper entitled “Specialized metabolite profiling between Cordyceps sinensis and confounders by untargeted UHPLC-MS/MS”

The paper was interesting to read although there are some typical errors and questions/clarifications below to better understand the work done by authors.

Some typical errors

Line 16:  Kindly write LC-MS instead of LS-MS

Line 105: “Took a 50 mg”  

Line 108: The sample solution was ground by by the….

Line 121:  ultra - - high performance…

Line 126: water(solvent)  Be consistent with space.          

Line 169: “Cloud platform”  instead of  “ Choud platform”.

Line 415: Kindly write LC-MS/MS instead of LS-MS/MS.     

………………………………………….

Some questions / clarifications:

11-     Why did you choose the title with UHPLC-MS/MS if along the paper it is not possible to see the fragments of your compounds.

22-      Why did you change the method using the positive/negative ion mode separation?

33-     Why did you choose a range of 70-1050 m/z?

44-     Why are you calling these metabolites “supercategories”?

55-     Line 19: The 18….differences, involving lipids and lipid like molecules….? What do you want to explain exactly? Please rephrase.

66-     What do you understand by “biological “ replicate?

77-     Line 97: Can you explain the “6 biological replicates”?

88-     From lines 95 to 99 : Kindly rephrase that sentence.

99-     Line 100: What was your aim while selecting those solvents?

110- Be clear please. From line 104 to 105. Did you grind wet sample and obtain your solid powder?

111- What is the aim of adding a steel bull in this experiment and why did you choose 6 mm instead of 3 mm for example?

112- Line 120: The LC-MS method you are using, is it yours or a modified one. Please give the reference.

113- Line 247:  I believe that C.sinensis was collected the same day and used for analyses. Which explanation can you give for this change of values of YS?

114-   Kindly check for more papers on C.sinensis and justify your results.

115-  Lines 404 to 406: According to your paper, how a consumer can effectively distinguish C.sinensis from similar products?

116- Line 415: Where you also looking for the fragments during this work? If yes where are their results?

References

Please be consistent while writing your references. Kindly correct these lines and check the remaining.

Lines 447, 448, 470, 475, 476, 508, 515, 518, 525, 539…

Author Response

Some typical errors

Line 16: Kindly write LC-MS instead of LS-MS

Line 105: “Took a 50 mg”

Line 108: The sample solution was ground by by the….

Line 121: ultra--high performance…

Line 126: water(solvent)  Be consistent with space.

Line 169: “Cloud platform” instead of “Choud platform”.

Line 415: Kindly write LC-MS/MS instead of LS-MS/MS.

Thank you so much for your careful check.We have modified the spelling errors,

Line 27 and Line 39: LC-MS

Line 123: Then 50 mg solid sample was added in a 2 mL centrifuge tube

Line126: The sample solution was ground by the

Line140: ultra­high performance liquid chromatography system

Line145: water (solvent A)

Line190: majorbio choud platform (cloud.majorbio.com)

Line450: LC-MS/MS

Comments 1: Why did you choose the title with UHPLC-MS/MS if along the paper it is not possible to see the fragments of your compounds.

Response 1: Thanks for your comments. This experiment utilized the ultra-high performance liquid chromatography tandem Fourier transform mass spectrometry (UHPLC-Q Exactive HF-X) system from Meiji Biotech for LC-MS/MS analysis, also known as UHPLC-MS/MS. We have revised the title 2.4 of the Materials and Methods section to “UHPLC-MS/MS analysis”. (See in L139)

Comments 2: Why did you change the method using the positive/negative ion mode separation?

Response 2: Thank you for pointing this out. In mass spectrometry analysis, positive and negative mode separation is used to better detect and distinguish metabolites. The positive and negative ion mode means that after the sample is ionized in the electric spray ionization (ESI) source, the ions with positive charges and negative charges will appear at the same time. Due to the differences in physicochemical properties of different metabolites, some metabolites may carry a positive charge, while others may carry a negative charge. By scanning these two ion states separately, more comprehensive metabolomic information can be obtained. This method helps to improve the sensitivity and coverage of metabolite detection, thereby more accurately reflecting metabolic changes within the organism. The feasibility of this method can be demonstrated in other articles (Jia et al. 2024, Yang et al. 2024 and Sun et al. 2023). (See in L146 to L150)

Comments 3: Why did you choose a range of 70-1050 m/z?

Response 3: Thank you for your rigorous . In metabolomics, the mass range of 70-1050 m/z is chosen to cover the mass to charge ratio (m/z) of most small molecule metabolites. Metabolites are usually small molecules with a molecular weight of less than 1000 Da, and this mass range can effectively capture the mass spectrometry signals of these metabolites. By setting this range, it can ensure that most of the metabolites of interest are detected, thereby improving the comprehensiveness and accuracy of the research. The feasibility of this method can be demonstrated in other articles (Jia et al. 2024, Yang et al. 2024 and Sun et al. 2023). (See in L161)

Comments 4: Why are you calling these metabolites “supercategories”?

Response 4: considerationThank you so much for your careful check. We have changed “supercategories” to “superclass” to emphasize this point. “Supercategories” refers to a broader category or category that includes multiple different categories or subcategories. The metabolite classification in the HMDB database generally uses “Kingdom”, “Superclass”, “Class”, and “Subclass”. So I'm sorry for my mistake in using words in the article. (See in L30)

Comments 5: Line 19: The 18….differences, involving lipids and lipid like molecules….? What do you want to explain exactly? Please rephrase.

Response 5: Thank you for your rigorous consideration. We have rephrased “The 18 superclasses were found to have differences, involving lipids, organic acids, nucleosides, carbohydrates, amino acids, vitamins, and their derivatives.” to emphasize this point. (See in L30 to L31)

Comments 6: What do you understand by “biological replicate”?

Response 6: Thank you for your rigorous consideration. “Replicates” are repeated measurements of the same sample using instruments, solely considering the reliability of the instrument. The sample for “Repeats” should be directly derived from the population of the research subject and independently repeated multiple times for all experiments. (See in L122)

Comments 7: Line 97: Can you explain the “6 biological replicates”?

Response 7: Thank you for your rigorous consideration. We have changed “6 biological replicates” to “6 biological repeats” to emphasize this point, which enhances the reliability of experimental data results. (See in L122)

Comments 8: From lines 95 to 99 : Kindly rephrase that sentence.

Response 8: Thank you for your rigorous consideration. I have rephrase the sentence. “C. sinensis, C. liangshanensis, C. gracilis, C. hawkesii, and C. gunnii were collected and identified authentic strains by Professor Li Yuling's Cordyceps research team, which are stored in the Microbial Center strain storage library. The in different producing areas of mature Cordyceps were shown in Fig. 1. The research in 2024 (Wang et al. 2024) and 2023 (Tang et al. 2023) mentioned the collection and preservation of Cordyceps by Li Yulin's team. (See in L114 to L117)

Comments 9: Line 100: What was your aim while selecting those solvents?

Response 9: Thank you for your rigorous consideration. In metabolomics research, selecting appropriate solvents is crucial for effectively extracting metabolites. Methanol and acetonitrile are commonly used organic solvents that can effectively precipitate proteins and extract metabolites. Formic acid can inhibit the degradation of nucleoside triphosphate into nucleosides and bases, protecting metabolites from destruction. Pure water is used for dilution or cleaning, while propanol helps extract certain specific metabolites. The combination of these solvents can improve the efficiency and coverage of metabolite extraction, ensuring the extraction of as many metabolites as possible from biological samples and providing a reliable data foundation for subsequent analysis. The feasibility of this method can be demonstrated in other articles (Jia et al. 2024, Yang et al. 2024 and Sun et al. 2023). (See in L118)

Comments 10: Be clear please. From line 104 to 105. Did you grind wet sample and obtain your solid powder?

Response 10: Thank you for your rigorous consideration. First, clean the samples thoroughly and aired dry it. Weigh 50mg of the sample and place it in a centrifuge tube, then use steel balls to assist in grinding it into powder. We have rephrased “The samples were cleaned with distilled water and aired dry them. Then 50 mg solid sample was added in a 2 mL centrifuge tube from each Cordyceps sample and added a steel ball of 6 mm.”. (See in L122)

Comments 11: What is the aim of adding a steel bull in this experiment and why did you choose 6 mm instead of 3 mm for example?

Response 11: Thank you for your rigorous consideration. The main purpose of adding steel balls during metabolite extraction is to physically break down tissues and release metabolites within the tissues. The size of the steel ball has a significant impact on the crushing effect. Compared to 3mm steel balls, 6mm steel balls can provide greater impact force in tissue crushers, thereby more effectively breaking cell walls and releasing more metabolites. This physical fragmentation method helps improve the efficiency of metabolite extraction, ensuring the accuracy and reliability of subsequent analysis. The feasibility of this method can be demonstrated in other articles (Jia et al. 2024, Yang et al. 2024 and Sun et al. 2023). (See in L124)

Comments 12: Line 120: The LC-MS method you are using, is it yours or a modified one. Please give the reference.

Response 12: Thank you for your rigorous consideration. The LC-MS method was tested and innovated to a certain extent by Majorbio Bio Pharm Technology Co., Ltd. (Shanghai, China). This method integrates other LC-MS/MS methods and has been improved accordingly. It has been mentioned in lines 133 and 180 of the “Materials and Methods” section of the article. Meanwhile, the reliability of this method has been validated in other literature (Tang et al, 2023, Jia et al. 2024, Yang et al. 2024 and Sun et al. 2023). (See in L131 and L190)

Comments 13: Line 247: I believe thatC.sinensis was collected the same day and used for analyses. Which explanation can you give for this change of values of YS?

Response 13: Thank you for pointing this out. We believe that the main reason is the difference in geographical environment. Compared to other regions, Qinghai has a higher altitude, longer sunshine hours, larger temperature differences, and less precipitation. This may be the reason for the differences in metabolites between C. sinensis and other Cordyceps. Studies have shown that drying significantly affects the metabolite content of Cordyceps, with significant changes in amino acid metabolism. We have added this issue in lines 428 to 439 of the text to emphasize this point.(See in L422 to L433)

Comments 14: Kindly check for more papers on C.sinensis and justify your results.

Response 14: Thank you for pointing this out. We have added more papers in my article.

  1. 34. Kim, H.K.; Verpoorte, R. Sample Preparation for Plant Metabolomics. Anal. 2010, 21, 4–13.
  2. 40. Jin,Y.;Meng, X.; Qiu, Z.; Su, Y.; Yu, P.; Qu, P. Anti-tumor and anti-metastatic roles of cordycepin, one bioactive compound of Cordyceps militaris. Saudi. J. Biol. Sci. 2018, 25(5):991-995.

41 Wang, T.; Tang, C;, Xiao, M.; Cao, Z.; He, H.; He, M.; Li, Y.; Li, X. (2024). Analysis of metabolic spectrum characteristics of naturally and cultivated Ophiocordyceps sinensis based on non-targeted metabolomics. Sci. Rep. 2024, 14(1), 17425.

42 Xiao, J.H.; Qi, Y.; Xiong, Q. Nucleosides a valuable chemical marker for quality control in traditional Chinese medicine Cordyceps. Recent Pat. Biotechnol. 2013.

50 Carturan, L.; De, B.F.; Dinale, R.; Dragà, G.; Gabrielli, P.; Mair, V.; Seppi, R.; Tonidandel, D.; Zanoner, T.; Zendrini, T.L.; et al.Modern air, englacial and permafrost temperatures at high altitude on Mt Ortles (3905m a.s.l.), in the eastern European Alps. Earth Syst. Sci. Data. 2023, 15, 4661–4688.

51 StanisÅ‚aw, B.; Mariusz, C.; Ulf, B.; Edyta, T.S. High oxidative stress despite low energy metabolism and vice versa: Insightsthrough temperature acclimation in an ectotherm. J. Therm. Biol. 2018, 78, 36–41.(See inL353, L387, L388, L432 and L433)

Comments 15: Lines 404 to 406: According to your paper, how a consumer can effectively distinguish C.sinensis from similar products?

Response 15: Thank you very much for pointing out this important issue. We have added “Using untargeted LC-MS/MS, we found that the content and types of lipids, nucleosides, and other compounds in C. sinensis were significantly higher than other Cordyceps,” to emphasize this point. At the same time, we have also added this deficiency to the limitation section, and we will collect targeted metabolomics and transcriptomics for further study in the future. Our current research indicates that eight biomarkers in the conclusion section can provide reference for consumers to distinguish C. sinensis and its confused products through metabolites. In addition, we also determine the content and types of organic compounds. In our study, the content and types of lipids and nucleoside compounds in C. sinensis were significantly higher than those in other Cordyceps, a viewpoint that has been confirmed in other literature. (See in L435 to L438, L443 to L448, and L460 to L462)

Comments 16: Line 415: Where you also looking for the fragments during this work? If yes where are their results?

Response 16: Thank you for pointing this out. We have added the fragments during this work to emphasize this point. “There were significant differences in 8 biomarkers, including Zedaorol, hylolycin B, P-Ctlvc-T, 6a-hydrox-ypaclitaxel, between C. sinensis and other alternatives. C. sinensis is rich in hydroxyinosine and is a potential HIV treatment drug.” Meanwhile, the result is mentioned in lines 370 to 385 of the discussion section. (See in L456 to L459, and L460 to L462)

Please be consistent while writing your references. Kindly correct these lines and check the remaining.

Lines 447, 448, 470, 475, 476, 508, 515, 518, 525, 539…

Thank you so much for your careful check. We have corrected the formatting errors of these references.

Reviewer 2 Report

Comments and Suggestions for Authors

1. Title- The title may be rephrased, and words like ‘’specialized and confounders may be deleted.

2.   Abstract- The abstract is relatively well written summarizing key aspects of the research. Authors should state in full at first mention less known abbreviations like KEGG.

3.    Introduction- The authors attempted to provide a good background for the study. The motivation for the study is relatively good. The authors should expand the information on the global medicinal uses of the Cordyceps spp. All scientific names should be italicized.

4.     Materials and Methods- In section 2.1, the Authors should rephrase the statement “The specimens of mature Cordyceps were shown in Fig. 1. “. The authors should indicate how the Cordyceps were identified and if the specimens were curated.

5.     Results- The results were relatively well presented. Authors should state in full at first mention all abbreviations.

6.     Discussion. The discussion was well written, with relevant comparisons with past studies done. Generally relevant inferences were excellently made.

7.  Conclusion. The authors gave good concluding remarks. Future perspectives were not provided.

8.       References. Most cited references were current and relevant.

9.  The authors should check the manuscript to correct grammar errors, mainly in the Introduction and discussion sections. 

Author Response

Comments 1: The title may be rephrased, and words like ‘’specialized and confounders may be deleted.

Response 1: We appreciate your so carefully reviewing for the manuscript. We have taken your advice and made changes to the title based on the content of the article. Change the title to “Comparative Metabolite Profiling between Cordyceps sinensis and other Cordyceps by Untargeted UHPLC-MS/MS” (See in L2)

Comments 2: The abstract is relatively well written summarizing key aspects of the research. Authors should state in full at first mention less known abbreviations like KEGG.

Response 2: Thank you for pointing this out. We have added “Kyoto Encyclopedia of Genes and Genomes (KEGG)” to emphasize this point.(See in L35)

Comments 3: The authors attempted to provide a good background for the study. The motivation for the study is relatively good. The authors should expand the information on the global medicinal uses of the Cordyceps spp. All scientific names should be italicized.

Response 3: A very nice suggestion, thanks a lot. We have added examples of the global medicinal and health value of Cordyceps. “Cordyceps is used as an energy supplement for athletes. The research team found through studying mice that Cordyceps helps to clear lactate and enhance anaerobic respiration in mouse cells. On the other hand, energy and improving internal mechanisms can increase the concentration of cellular bioenergy ATP, thereby improving the efficiency of oxygen utilization and enabling athletes to compete in oxygen scarce conditions.”(See in L82 to L87)

Comments 4: In section 2.1, the Authors should rephrase the statement “The specimens of mature Cordyceps were shown in Fig. 1. “. The authors should indicate how the Cordyceps were identified and if the specimens were curated.

Response 4: Thanks your suggestion. We have rephrased this paragraph. “C. sinensis, C. liangshanensis, C. gracilis, C. hawkesii, and C. gunnii were collected and identified authentic strains by Professor Li Yuling's Cordyceps research team, which are stored in the Microbial Center strain storage library. The in different producing areas of mature Cordyceps were shown in Fig. 1.” The research in 2024 (Wang et al. 2024) and 2023 (Tang et al. 2023) mentioned the collection and preservation of cordyceps by Li Yulin's team. (See in L114 to L117)

Comments 5: The results were relatively well presented. Authors should state in full at first mention all abbreviations.

Response 5: Thank you very much for raising the question. We have added “We use UHPLC-Q Exactive HF-X untargeted metabolomics to obtain characteristic differential metabolites of C. gracilis(XJ), C. sinensis(YS), C. hawkesii(YX), C. liangshanensis(LS) and C. gunnii(GN)” at the beginning of the results section.(See in L193 to L195)

Comments 6: The authors gave good concluding remarks. Future perspectives were not provided.

Response 6: A very nice suggestion, thanks a lot. We have added “Future efforts will focus on targeted metabolomics and transcriptomic analyses of these differential metabolites to further validate and quantify their production processes and pharmacological effects.Provide theoretical basis for the development and utilization of pharmacological activities of cordyceps and its substitutes”, which can treat uterine bleeding, but improper use can lead to ergot poisoning on lines 464, page 13 to emphasize this point. (See in L464 to L468)

Point: The authors should check the manuscript to correct grammar errors, mainly in the Introduction and discussion sections.

Response: Thank you so much for your careful check. The article has been edited in the native English speaker, and all the spelling errors have been corrected.
